# Morphometric and molecular discrimination of the sugarcane aphid, *Melanaphis sacchari*, (Zehntner, 1897) and the sorghum aphid *Melanaphis sorghi* (Theobald, 1904)

Samuel Nibouche[1]*, Laurent Costet[1], Raul F. Medina[2], Jocelyn R. Holt[2], Joëlle Sadeyen[3], Anne-Sophie Zoogones[1,3], Paul Brown[4], Roger L. Blackman[4]

**1** CIRAD, UMR PVBMT, Saint Pierre, La Réunion, France, **2** Texas A&M University, College Station, Texas, United States of America, **3** UMR PVBMT, Université de La Réunion, Saint Pierre, La Réunion, France, **4** The Natural History Museum, London, United Kingdom

* samuel.nibouche@cirad.fr

**Data Availability Statement:** All sequence data are available in Genbank, and the accession numbers are listed in S1 Table. Raw morphometric data are

## Abstract

*Melanaphis sacchari* (Zehntner, 1897) and *Melanaphis sorghi* (Theobald, 1904) are major worldwide crop pests causing direct feeding damage on sorghum and transmitting viruses to sugarcane. It is common in the scientific literature to consider these two species as synonyms, referred to as the 'sugarcane aphid', although no formal study has validated this synonymy. In this study, based on the comparison of samples collected from their whole distribution area, we use both morphometric and molecular data to better characterize the discrimination between *M. sacchari* and *M. sorghi*. An unsupervised multivariate analysis of morphometric data clearly confirmed the separation of the two species. The best discriminating characters separating these species were length of the antenna processus terminalis relative to length of hind tibia, siphunculus or cauda. However, those criteria sometimes do not allow an unambiguous identification. Bayesian clustering based on microsatellite data delimited two clusters, which corresponded to the morphological species separation. The DNA sequencing of three nuclear and three mitochondrial regions revealed slight divergence between species. In particular, the COI barcode region proved to be uninformative for species separation because one haplotype is shared by both species. In contrast, one SNP located on the nuclear EF1-α gene was diagnostic for species separation. Based on morphological and molecular evidence, the invasive genotype damaging to sorghum in the US, Mexico and the Caribbean since 2013 is found to be *M. sorghi*.

## Introduction

The species *Melanaphis sacchari* [1] and *Melanaphis sorghi* [2] were described at the turn of the 20[th] century by Zehntner [1] on sugarcane in Java and by Theobald [2] on sorghum in Sudan respectively. However, although these two species are commonly treated as synonyms, referred to as the 'sugarcane aphid', no comparative study demonstrating this synonymy has

available at https://doi.org/10.18167/DVN1/PDPDS4. R and SAS codes are available on Github (https://github.com/SamNibouche/Melanaphis_taxonomy). Specimen metadata are available on the Arthemis website (http://arthemisdb.supagro.inra.fr/). The authors had no special access privileges, and other researchers will be able to access this data in the same manner as the authors.

**Funding:** This work was co-funded by the European Union: Agricultural Fund for Rural Development (https://ec.europa.eu/agriculture/cap-funding_en), by the Conseil Départemental de La Réunion (http://www.cg974.fr), by the Conseil Régional de La Réunion (https://www.regionreunion.com) and by the Centre de Coopération internationale en Recherche agronomique pour le Développement (www.cirad.fr), as well as by the National Institute of Food and Agriculture, U.S. Department of Agriculture Hatch Program (TEX09185 to R. F. M), and was carried out in part on the Plant Protection Platform which is co-financed by the Groupe d'Intérêt Scientifique "Infrastructures en Biologie Santé et Agronomie" (www.ibisa.net). The funders had no role in study design, data collection and analysis, decision to publish, or preparation of the manuscript.

**Competing interests:** The authors have declared that no competing interests exist.

been conducted. When Remaudière & Remaudière [3] considered *M. sorghi* as a synonym of *M. sacchari* in their 1997 catalogue, following Eastop (1953) [4], they provided no reference to support this choice. Moreover, Halbert and Remaudière [5] referred later to these species as the 'sorghi/sacchari group' and underlined them as '*two very variable species usually regarded as synonyms, but possibly distinct according to Blackman et al.* [6]'. The communication by Blackman et al. [6] indeed supported the separation of *M. sorghi* and *M. sacchari* and provided a morphological criterion to separate the species, based on the ratio between the hind tibia length and the antennal processus terminalis length. The same separation was used in Blackman and Eastop's book in 2006 [7] and has not been challenged since.

Margaritopoulos et al. [8] stated that '*DNA evidence that might confirm the existence of two species is not yet available, but at this time it seems advisable to recognise that they are probably functioning as distinct taxonomic entities*'. Nibouche et al. [9] observed genetic structuring in five clonal lineages matching a geographic structure, but they could not separate the two species by using 'universal' COI barcoding.

Regarding host plant association, each species is observed on both sugarcane and sorghum, but *M. sorghi* is considered preferring sorghum and *M. sacchari* preferring sugarcane [6]. However, host plant association is blurred by the existence of biotypes, as shown by Nibouche et al. [10] who demonstrated the existence of a sorghum and a sugarcane biotype in Reunion populations (within the same multi locus lineage). Interestingly, *Melanaphis sorghi* is known for a long time to produce very heavy infestations on sorghum, in Africa. Early in the 20th century, Vuillet & Vuillet (1914) [11] cited this aphid as responsible for famines in West Africa.

The objectives of this study were (1) to improve the description of the morphometric differences between *M. sorghi* and *M. sacchari*, and (2) to delimit the molecular separation of *M. sorghi* and *M. sacchari*.

## Material and methods

### Material collected

This study is based on 199 samples, collected from 2002 to 2016 in 31 states or countries (S1 Table). No specific permissions were required when sampling aphids in the locations studied. From these 199 samples, we analyzed 2,409 apterous aphid specimens collected on cultivated sorghum (*Sorghum bicolor*) (n = 439), Johnson grass (*Sorghum halepense*) (n = 97), sugarcane (*Saccharum officinarum* x *S. spontaneum*) (n = 1,382), *Sorghum arundinaceum* (≡ *Sorghum bicolor verticiliflorum* ≡ *Sorghum verticiliflorum*) (n = 427), maize (*Zea mays*) (n = 2), perennial sorghum (*Sorghum × almum*) (n = 8), *Sorghum* sp. (n = 16), and pearl millet (*Pennisetum glaucum*) (n = 38). Most of this material was already analyzed in previous studies [9, 10, 12]. Using Blackman & Eastop's (2006) key, we identified these specimens as *M. sacchari* or *M. sorghi*, but the identification sometimes was ambiguous given the continuous and overlapping nature of the criteria used to separate both species. For convenience, while awaiting this taxonomy study, this material was referred to as *M. sacchari* in our previous papers.

### Morphometry

**Material.** The morphometry dataset consisted of 89 apterous female specimens that were slide-fixed (S1 Fig) after a non-destructive DNA extraction. Among these 89 specimens, 21 were successfully genotyped with SSR and could be assigned to a multilocus lineage (MLL). The remaining 68 were not genotyped, but belonged to a sample whose multilocus lineage (MLL) had been identified from other specimens. Since 186 of the 188 (98.9%) SSR genotyped samples in this study were homogeneous (i.e. contained only a single MLL), we assumed that the non-genotyped slide-fixed samples belonged to the same MLL as the other specimens in

the sample that were genotyped. The distribution of the slide-fixed specimens was: 22 MLL-A, 2 MLL-B, 19 MLL-C, 15 MLL-D and 31 MLL-F. Since only two specimens of MLL-B were observed, they were discarded from the statistical analysis. No MLL-E specimen was observed.

We also examined the Theobald type series of *Aphis sacchari* collected in Sudan in 1902, which is stored in the Natural History Museum of London (NHM). In this series, only one paratype specimen (NHM-1915-81) was an apterous female and could be used for our morphometric analysis. Unfortunately, we were unable to locate the types of *Melanaphis sacchari*. Hollier & Hollier [13] reported that a fire destroyed the experimental station of Salatiga in Java in 1902, including Zehntner's laboratory and his collections. As Zehntner did not send his types to the Geneva museum, the type of *Melanaphis sacchari*, collected in 1897, probably was destroyed and should be considered lost.

We also examined some alate *M. sorghi* (n = 5) and *M. sacchari* (n = 5) specimens on slides from the MNHN (Muséum National d'Histoire Naturelle, Paris; G. Remaudière collection) and from the GBGP (Centre de Biologie et de Gestion des Populations, Montpellier; F. Leclant collection). The Theobald's type series from NHM contained one alate specimen (NHM-1915-81), but it was of insufficient quality to be included in the morphometric characterization of alatae.

**Methods.**   Twenty-two characters that are used classically in aphid taxonomy [7, 14] were measured on each slide-fixed specimen, using a binocular lens stereo microscope. Paired appendages (i.e. legs, antenna, siphunculi), were measured on both sides, and the mean value was used for analysis (except for the number of setae on the antenna, which was observed on one side only). Twelve ratios were computed from these characters.

The analysis was carried out on a subset of 11 characters (Table 1) that are known to be discriminant between species within the *Melanaphis* genus [5, 7, 14, 15]. Because of missing data, only 50 specimens were used in the multivariate analysis. Because Theobald did not clarify his *M. sorghi* paratype specimen (NHM-1915-81), we were unable to observe *urs* and *siphBW*, causing three missing ratios in the dataset. To include the paratype in the analysis, we replaced these three missing data by zeros after the standardization step (see below).

The 11 variables were standardized to obtain a mean of zero and a standard deviation of 1 prior to analysis.

The data were first submitted to a discriminant analysis of principal components (DAPC) with the R package ADEGENET using the *find.clusters* function [16]. This method first uses an unsupervised k-means clustering approach to determine the number of clusters without

**Table 1. List of the 11 morphological variables used in the discriminant analysis of principal components (DAPC) analysis.**

| Variable name | Variable signification |
|---|---|
| *NsetaeCauda* | number of setae on the cauda |
| *pt*:cauda | ratio processus terminalis length / cauda length |
| *HindTibia*:pt | ratio hind tibia length / processus terminalis length |
| *Ant*:BL | ratio antenna length / body length |
| *urs*:htII | ratio ultimate rostral segment length / hind tarsa II length |
| *pt*:VIb | ratio processus terminalis length / base length of the 6th antennal segment |
| *pt*:siph | ratio processus terminalis length / siphunculi length |
| *cauda*:urs | ratio cauda length / ultimate rostral segment length |
| *siph*:BL | ratio siphunculi length / body length |
| *siph*:siphBW | ratio siphunculi length / siphunculi basal width |
| *siph*:cauda | ratio siphunculi length / cauda length |

requiring any a priori clustering information. The determination of the number of clusters was based on the Bayesian Information Criteria (BIC). Then in a second step, a principal component analysis (PCA) is carried out, followed by a canonical discriminant analysis (CDA) performed on the coordinates along the principal components. To verify if the clustering was congruent with the separation of *M. sorghi* and *M. sacchari*, we compared the *HindTibia:pt* ratio between the clusters delimited by ADEGENET. Indeed, *M. sorghi* specimens have a relative length of the processus terminalis shorter than *M. sacchari* specimens [6, 7]. According to [7, 8, 14], the *HindTibia:pt* ratio range for apterae is (2.0–3.0) for *M. sorghi* vs. (1.4–2.2) for *M. sacchari*.

Secondly, the complete morphological dataset (i.e. 34 morphological traits on 88 apterous specimens) was submitted to a one-way ANOVA with SAS PROC GLM [17] to detect significant differences between *M. sacchari* and *M. sorghi*. Because we carried out multiple analysis, to control the study-wise type-1 error level we used a 5% False Discovery Rate (FDR) approach [18] with SAS PROC MULTTEST [17] to detect significant differences between clusters. To carry out this analysis, slide specimens were assigned to *M. sacchari* or *M. sorghi* according to their molecular assignment using SSR and EF1-α data.

## DNA extraction

DNA of individual aphids was extracted using the 'salting-out' protocol of Sunnucks and Hales [19] or using the Qiagen DNeasy Blood & Tissue Kit (Qiagen, Courtaboeuf, France). For slide-fixed specimens, a non-destructive DNA extraction was performed using the Qiagen manufacturer's protocol, but retrieving the insect body from the first elution column [12].

## Microsatellites

Nine microsatellite (Single Sequence Repeat, SSR) markers were selected among 14 previously developed markers [20]. PCR reactions were performed with labelled primers and multiplexed following previously established protocols [9]. Genotyping was carried out using an ABI PRISM 3110 and alleles were identified at each locus by comparison with a size standard using Gene-Mapper version 2.5 software (Applied Biosystems). The total microsatellite genotyping dataset included 2,255 specimens: 2,175 specimens were previously analyzed in [9, 10, 12] and 80 additional specimens were genotyped for this study. Single combinations of alleles were characterized and arranged as distinct multilocus genotypes (MLG) and assigned to one of six multilocus lineages (MLL) [9, 12]. We carried out a Bayesian clustering analysis with Structure version 2.3.4 [21], results were summarized with Structure Harvester [22], Clumpp [23] and Distruct 1.1 [24]. Parameters of the Structure analysis were: admixture, independent allele frequencies, 100,000 iterations after a 25,000 burn-in period, 10 replications for each k value ranging from 1 to 8.

## DNA sequencing

Aphids were sequenced for three mitochondrial and three nuclear DNA regions belonging to the cytochrome c oxidase subunit I (COI) [25], *c*ytochrome c oxidase II (COII) [26], cytochrome b (CytB) [27], elongation factor-1α (EF1-α) [28] and the internal transcribed spacer 1 and 2 (ITS1 and ITS2) [29].

Three of the six sequences, COI, COII and EF1-α, produced informative polymorphism and were simultaneously sequenced on a large number of specimens. These sequences were concatenated and the resulting concatenated haplotypes were used to produce a minimum spanning network with PopArt [30].

## Results

### Morphometry

According to the unsupervised k-means clustering, the number of inferred morphological clusters was k = 2. The membership probability of each specimen is shown by Fig 1. The *M. sorghi* paratype was assigned to the morphological cluster 1, with a 100% membership probability.

Both clusters differed significantly (P < 0.0001) by their *HindTibia*:*pt* ratio. The blue cluster (Fig 1), which contained the *M. sorghi* paratype and exhibited the lowest *HindTibia*:*pt* (Fig 2 and Table 2), was *M. sorghi*. The orange cluster (Fig 1), which exhibited the largest *HindTibia*:*pt* ratio (Fig 2 and Table 2), corresponded to *M. sacchari*.

The loading plot (Fig 3) showed that the morphological characters contributing most to the DAPC were *pt*:*cauda*, *HindTibia*:*pt* and *pt*:*siph*. The graphical comparison of these three main contributing traits between both species is presented in Fig 2. In these plots, the assignment of the specimens to both species was made according to their SSR and EF1-α genotype (see below). For the three traits, both species differ significantly with an uncorrected P-value < 0.0001 (F = 100.3, 78.73 and 62.96 respectively for *pt*:*cauda*, *HindTibia*:*pt* and *pt*:*siph*). However, some overlapping is observed for each trait.

The comparison of all morphological traits of *M. sorghi* and *M. sacchari* apterous females is presented in Table 2. There are significant differences between both species in 16 out of 34 traits.

The values for alatae are given in S2 Table. Due to the small number of specimens observed, no statistical comparison was carried out and these values are only suggestive. Because we only observed museum slide-fixed alate specimens, we could not genotype them. In the absence of a key for alatae, species assignment was carried out taking into account geographic origin, according to the worldwide repartition of both species (Fig 6). Specimens from West Africa (Burkina Faso, Ivory Coast and Senegal) were considered as *M. sorghi*, specimens from Reunion were considered as *M. sacchari*. In Brazil, we assumed that the recent detection of *M. sorghi* (Fig 6) was due to the ongoing invasion of the Americas by this species (see below) and therefore the alate specimen we observed, which was collected in 1968, was *M. sacchari*.

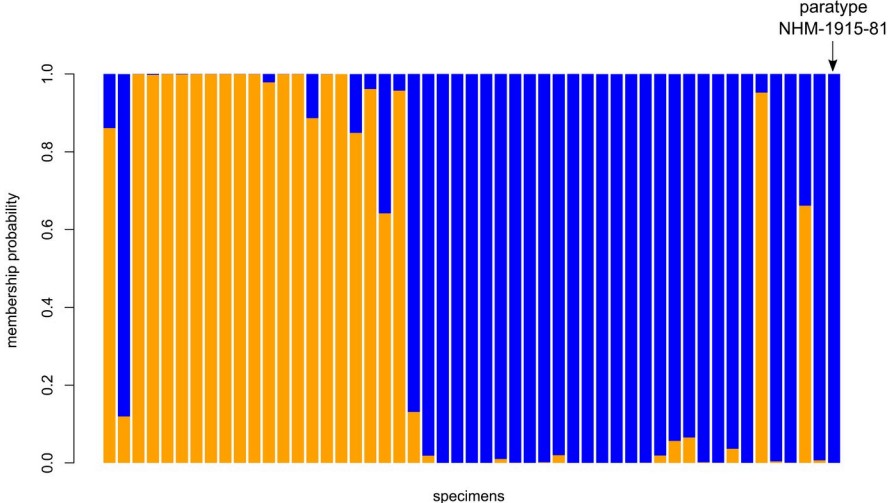

**Fig 1. DAPC analysis based on 11 morphological traits recorded on 51 slide-fixed specimens.** Species identification by SSR and EF1-α: orange = *M. sacchari*, blue = *M. sorghi*.

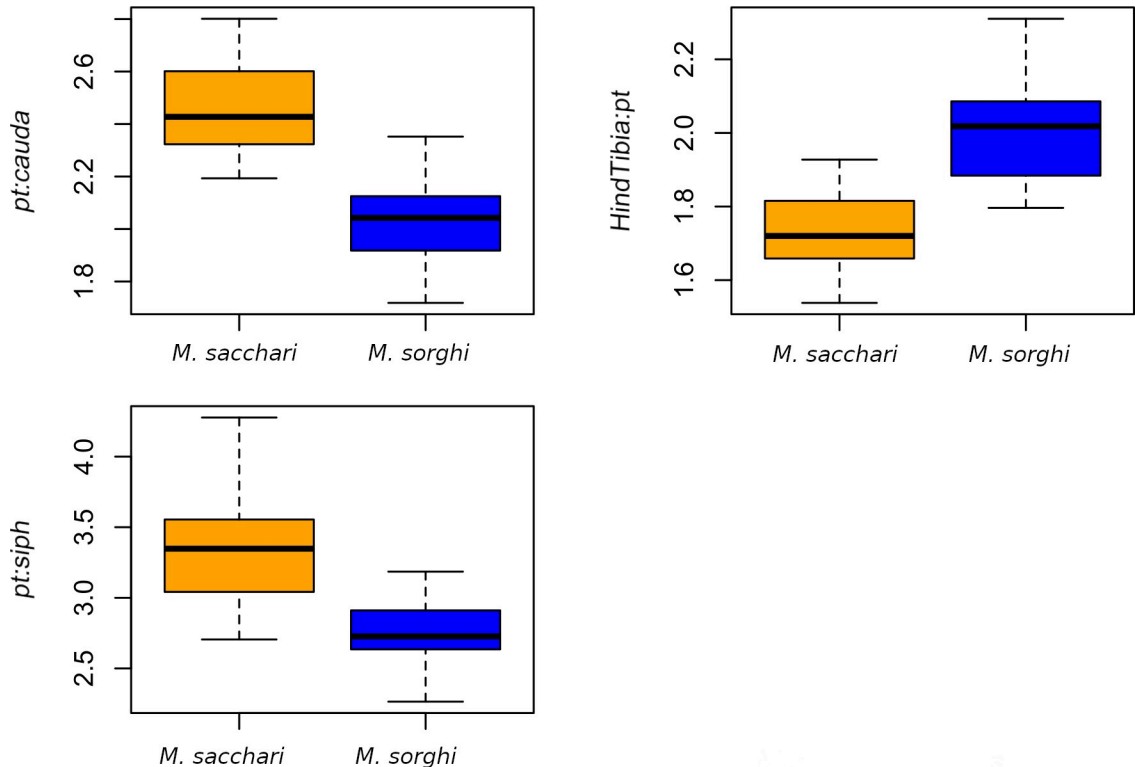

**Fig 2. Comparison between *M. sacchari* and *M. sorghi* using the three traits showing the highest loadings in the DAPC among 51 slide-fixed specimens.** The specimens are assigned to *M. sorghi* or *M. sacchari* according to their SSR or EF1-α genotype.

Although the number of alate specimens is insufficient to form a basis for any statistical comparison, it should be noted that, as in apterae, the *pt* of alatae assigned to *M. sacchari* tends to be longer than that of alatae assigned to *M. sorghi*, and that the ratios involving this parameter, especially *pt:siph*, may prove to be useful discriminants.

## Microsatellites

Fifty-nine multilocus genotypes (MLG) have been identified (S6 Table). Fifty-six were published earlier [9, 10, 12] and three new ones (Ms25, Ms26, and Ms58) were observed during this study. The clustering with Structure, followed by the use of the Evanno et al. [31] method, leads to the conclusion that the number of inferred populations is k = 2 (S2 Fig).

The assignment of each MLG with k = 2 is presented in Fig 4. The first cluster groups MLL-A-E-F, the second cluster groups MLL-B-C-D.

The congruence between the DAPC morphological assignment to *M. sorghi* or *M. sacchari* and the Structure Bayesian clustering based on SSR genotyping showed unambiguously that the blue cluster is *M. sorghi* and the orange cluster is *M. sacchari* (Table 3). Among the 29 specimens belonging to the blue Structure cluster, 27 were morphologically assigned to *M. sorghi*, one was morphologically assigned to *M. sacchari* and one was undetermined. Among the 21 specimens belonging to the orange Structure cluster, 19 were morphologically assigned to *M. sacchari*, one was morphologically assigned to *M. sorghi* and one undetermined. The resulting accuracy of the congruence of Structure clustering and morphological clustering is 95.8% (46 / 48), when excluding the two undetermined specimens.

**Table 2. Comparison of morphological characteristics of *M. sacchari* and *M. sorghi* apterous females (mean values, with range under brackets).**

| | *M. sacchari* | | *M. sorghi* | | P-value (FDR) |
|---|---|---|---|---|---|
| | **n = 34 (16 samples)** | | **n = 54 (29 samples)** | | |
| BL | 1338 | (1060–1696) | 1414 | (1013–1811) | 0.2104 |
| cauda | 126 | (96–151) | 147 | (103–180) | <0.0001 |
| caudaBW | 79 | (40–108) | 83 | (45–121) | 0.3837 |
| urs | 75 | (63–83) | 79 | (66–91) | 0.0069 |
| NsetaeCauda | 11.8 | (5–16) | 10.3 | (4–15) | 0.0319 |
| htII | 72 | (56–83) | 72 | (55–85) | 0.6388 |
| HindFemur | 317 | (242–413) | 350 | (254–421) | 0.0005 |
| HindTibia | 530 | (413–628) | 585 | (419–707) | 0.0002 |
| HindTibiaW | 35 | (27–48) | 36 | (29–53) | 0.3837 |
| siph | 95 | (67–123) | 108 | (77–130) | 0.0002 |
| siphDW | 38 | (30–47) | 42 | (36–49) | <0.0001 |
| siphBW | 60 | (43–76) | 62 | (44–88) | 0.3837 |
| AntI | 57 | (48–69) | 61 | (49–80) | 0.0121 |
| AntII | 44 | (35–58) | 49 | (40–64) | 0.0019 |
| AntIII_IV | 293 | (207–374) | 322 | (226–413) | 0.0110 |
| AntV | 146 | (115–181) | 144 | (101–184) | 0.7456 |
| VIb | 79 | (69–90) | 80 | (65–95) | 0.8727 |
| pt | 308 | (258–359) | 294 | (207–345) | 0.0568 |
| AntIIIBW | 23 | (19–30) | 24 | (18–30) | 0.3775 |
| NsetaeAntIII_IV | 4.2 | (2–7) | 5.0 | (2–9) | 0.0568 |
| NsetaeAntV | 2.3 | (1–3) | 2.4 | (1–4) | 0.7108 |
| Ant | 935 | (754–1101) | 949 | (695–1156) | 0.6196 |
| Ant:BL | 0.72 | (0.6–0.85) | 0.69 | (0.59–0.86) | 0.2434 |
| urs:htII | 1.05 | (0.9–1.26) | 1.08 | (0.83–1.34) | 0.3837 |
| pt:VIb | 3.87 | (3.3–4.38) | 3.70 | (3.1–4.3) | 0.0311 |
| pt:cauda | 2.46 | (2.19–2.8) | 2.02 | (1.72–2.35) | <0.0001 |
| pt:siph | 3.33 | (2.71–4.28) | 2.75 | (2.26–3.19) | <0.0001 |
| HindTibia:pt | 1.73 | (1.54–1.93) | 2.00 | (1.8–2.31) | <0.0001 |
| cauda:urs | 1.69 | (1.34–1.95) | 1.85 | (1.42–2.2) | 0.0003 |
| urs:VIb | 0.94 | (0.83–1.11) | 0.99 | (0.81–1.18) | 0.0776 |
| siph:BL | 0.07 | (0.06–0.09) | 0.08 | (0.06–0.1) | 0.1304 |
| siph:siphBW | 1.61 | (1.10–2.12) | 1.77 | (1.30–2.87) | 0.0311 |
| siph:cauda | 0.75 | (0.59–0.89) | 0.74 | (0.61–0.93) | 0.3837 |
| cauda:caudaBW | 1.66 | (1.16–3.06) | 1.82 | (1.17–3.49) | 0.1711 |

Significant differences are highlighted in gray. The specimens are assigned to *M. sorghi* or *M. sacchari* according to their own SSR or EF1-α genotype or according to the genotype of the sample they belong to. P-values are corrected for multiple testing by the False Discovery Rate method. Measures are in μm or unitless. n = maximum number of specimens observed, the actual number depends on the missing data within each trait.

## DNA sequencing

A total of 371 aphids were sequenced for at least one of three genes, COI, COII or EF1-α: 340 for COI, 143 for COII, and 163 for EF1-α.

In COI (658 bp), we detected four SNPs defining four haplotypes (S3 Table). The sequences were deposited in Genbank under accession numbers KJ083108-KJ083215, KX453783-KX453784, MG838208-MG838315, and MT813521-MT813656.

In COII (763 bp), we detected four SNPs defining five haplotypes (S4 Table). The sequences were deposited in Genbank under accession numbers MT847245-MT847387.

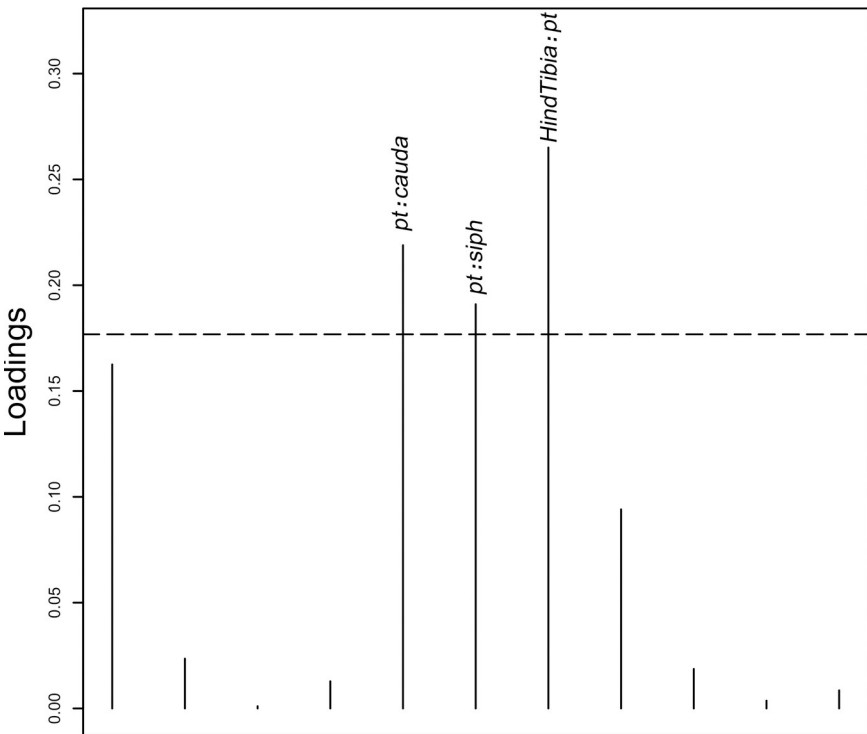

**Fig 3. Canonical loading plot.** The horizontal line is the limit showing the variables (morphological traits) that yield a cumulated 75% contribution to the DAPC. The individual peaks show the magnitude of the influence of each variable on separation of *M. sorghi* and *M. sacchari*.

The EF1-α gene portion amplified was 1,014 bp long. The sequences were deposited in Genbank under accession numbers MT847432-MT847594. Prior to analysis, the sequences

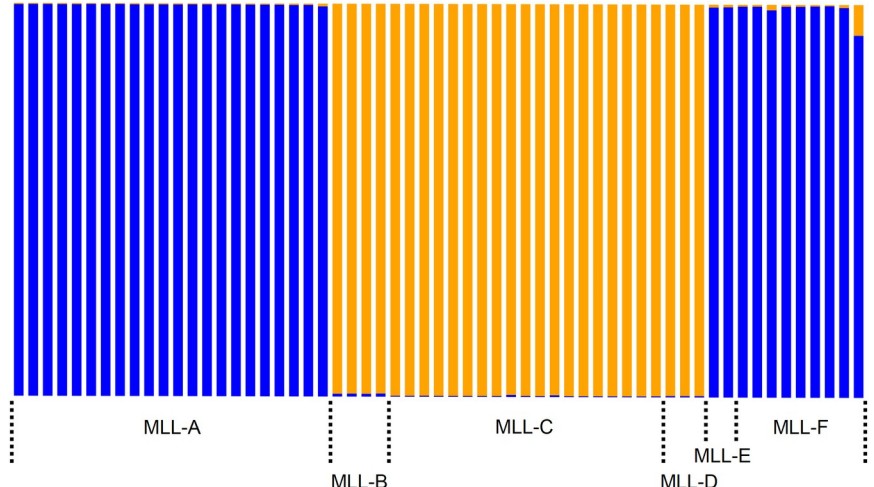

**Fig 4. Assignment of each of the 59 MLG to the two clusters inferred by Structure.** MLG were defined using nine microsatellite markers.

**Table 3. Multi Locus Lineage (MLL) identification of the 51 slide fixed specimens morphologically assigned to *M. sorghi* or *M. sacchari* by DAPC.**

| Structure cluster | MLL | level of MLL identification [b] | morphological assignment by DAPC | | |
|---|---|---|---|---|---|
| | | | *M. sacchari* | *M. sorghi* | undetermined [a] |
| orange | MLL-C | specimen | 1 | | 1 |
| | | sample | 9 | 1[c] | |
| | MLL-D | specimen | 8 | | |
| | | sample | 1 | | |
| blue | MLL-A | specimen | | | |
| | | sample | | 16 | |
| | MLL-F | specimen | 1[c] | 7 | |
| | | sample | | 4 | 1 |

[a] The specimens with a DAPC membership lower than 0.8 are considered undetermined.

[b] MLL identification was carried out by SSR genotyping on the specimen itself or on other specimens from the same sample.

[c] Specimens whose molecular and morphometric assignments are discordant.

were trimmed to a 528 bp length (from position 248 bp to 775 bp), to discard low quality 5' and 3' sequence ends in most specimens. In this EF1-α 528 bp sequence portion, we detected 1 unambiguous SNP and 4 ambiguous positions (heterozygous) defining seven haplotypes (S5 Table). Only two haplotypes were defined when omitting the four ambiguous positions, and we considered only these two haplotypes in further analysis. Haplotype H1 was only observed in *M. sacchari* and haplotype H2 only in *M. sorghi*.

In CytB (745 bp), we detected one SNP defining two haplotypes. Both haplotypes were detected in both *M. sacchari* and *M. sorghi*. The sequences were deposited in Genbank under accession numbers MT847388-MT847431.

In the ITS1 region, we obtained 445–451 bp length sequences. These sequences included a 3–6 bp indel region and one SNP. The two haplotypes defined by the SNP were present in both *M. sacchari* and *M. sorghi* specimens. The sequences were deposited in Genbank under accession numbers MT821305-MT821342.

In the ITS2 region, in a 462 bp-sequence length, we detected one SNP and three indels, but the SNP was located within an indel and therefore could not be used in further analysis. The sequences were deposited in Genbank under accession numbers MT821344-MT821448.

A total of 63 specimens were genotyped at the three genes COI, COII, EF1-α, and were also successfully genotyped with SSR. The relationship between MLL and haplotypes among these specimens is presented in Table 4. EF1-α provided a diagnostic substitution at position 637 separating *M. sorghi* from *M. sacchari*. The distinctive base was T for *M. sacchari* and A for *M. sorghi*. COI provides an incomplete separation of both species: haplotypes H2 and H3 are diagnostic of *M. sacchari*, H6 is diagnostic of *M. sorghi*, but haplotype H1 is present in both species.

The number of variable sites was nine on the 1,806 bp of the three concatenated gene sequences (i.e. 0.49%), defining eight haplotypes, with a nucleotide diversity of 0.13%. The minimum spanning network among haplotypes is shown by Fig 5. *M. sorghi* specimens defined a star shaped haplogroup centered on haplotype cH1. *M. sacchari* defined a less homogeneous haplogroup, each haplotype being separated from the other by two or three substitutions.

The within species and between species divergences for the six sequenced gene portions are summarized in Table 5.

**Table 4. Haplotypes defined by the concatenation of four genes and correspondence with the Multilocus Lineages (MLLs) defined using SSR.**

| concatenated haplotype | gene / haplotype | | | species | MLL | Structure Cluster | n |
|---|---|---|---|---|---|---|---|
| | CO1 | CO2 | EF | | | | |
| cH1 | H1 | H3 | H2 | *sorghi* | A | 1 | 18 |
| cH1 | H1 | H3 | H2 | *sorghi* | F | 1 | 8 |
| cH2 | H1 | H3 | H1 | *sacchari* | B | 2 | 2 |
| cH3 | H2 | H1 | H1 | *sacchari* | C | 2 | 11 |
| cH4 | H2 | H5 | H1 | *sacchari* | C | 2 | 5 |
| cH5 | H3 | H3 | H1 | *sacchari* | D | 2 | 10 |
| cH6 | H1 | H2 | H2 | *sorghi* | E | 1 | 6 |
| cH7 | H6 | H3 | H2 | *sorghi* | F | 1 | 1 |
| cH8 | H1 | H4 | H2 | *sorghi* | A | 1 | 2 |

## Geographical distribution of *M. sacchari* and *M. sorghi*

The geographical distribution of *M. sacchari* and *M. sorghi*, is presented in Fig 6 and is based on 2,332 genotyped specimens. Specimens belonging to MLL-A-E-F were assigned to *M. sorghi*, while specimens belonging to MLL-B-C-D were assigned to *M. sacchari*. In the absence of SSR genotyping, specimens bearing the EF1-α haplotype H1 or the COI haplotypes H2 or H3 were assigned to *M. sacchari* while specimens bearing EF1-α haplotype H2 were assigned to *M. sorghi* (Table 5). The remaining specimens (n = 24) were considered unidentified and therefore not taken into account in Fig 6. Only one unambiguous (i.e. allowing the species identification) data point from India was available from a public database (see Fig 6 legend). Most COI sequences stored in these public databases are either haplotypes that we did not encounter in our study or are H1 haplotype, which is uninformative because it is shared by *M. sorghi* and *M. sacchari*.

In West and in Southern Africa, *M. sorghi* is the sole detected species. In East Africa, *M. sacchari* has been detected in Kenya and Tanzania, and *M. sorghi* in Uganda and Kenya. In

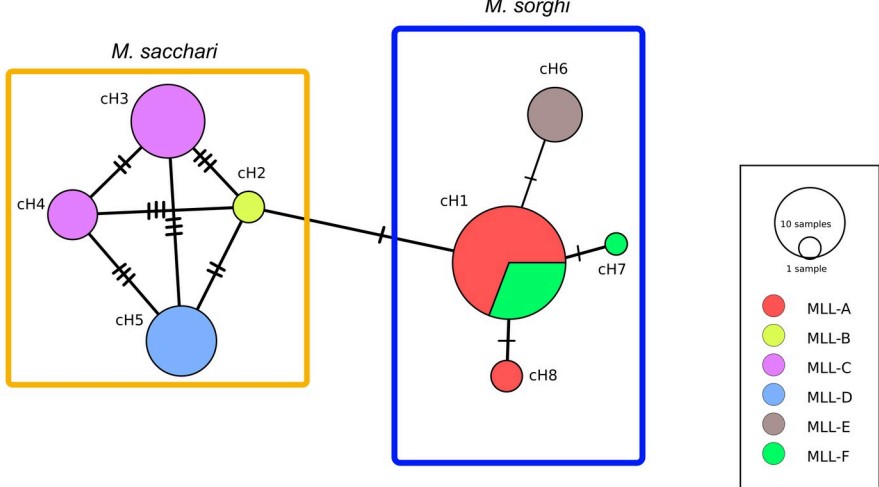

**Fig 5. Minimum spanning network constructed using the concatenated COI, COII and EF1-α sequences.** The orange and blue boxes indicate the STRUCTURE clusters inferred from SSR data (Table 5). The eight concatenated haplotypes (cH1 to cH8) are listed in Table 5. The number of hatch marks represents the number of mutations separating the concatenated haplotypes. Circle sizes are proportional to haplotype frequencies.

**Table 5. Sequence divergences (pairwise uncorrected P-distances, %) between or within species.**

|  | COI | COII | EF1α | CytB | ITS2 | ITS1 |
|---|---|---|---|---|---|---|
| within *M. sorghi* | 0.15 | 0.18 | 0 | 0 | 0 | 0.16 |
|  | (0.15–0.15) | (0.13–0.26) | - | - | - | (0–0.26) |
|  | n = 184 | n = 70 | n = 88 | n = 11 | n = 67 | n = 18 |
| within *M. sacchari* | 0.30 | 0.18 | 0 | 0.13 | 0 | 0.13 |
|  | (0.30–0.30) | (0.13–0.26) | - | (0.13–0.13) | - | (0–0.26) |
|  | n = 136 | n = 73 | n = 83 | n = 32 | n = 38 | n = 20 |
| between species | 0.28 | 0.18 | 0.19 | 0.09 | 0 | 0.13 |
|  | (0–0.45) | (0–0.26) | (0.19–0.19) | (0–0.13) | - | (0–0.26) |
|  | n = 320 | n = 143 | n = 171 | n = 43 | n = 105 | n = 38 |

Values are the mean of the pairwise distances; minimum and maximum distances for these comparisons are given in parentheses. Number of genotyped specimens = n. Species identification was carried out by SSR genotyping on the specimens themselves or on other specimens from the same sample.

Kenya, both species coexist and were collected in the same sample once. Reunion and Mauritius, in the South West Indian Ocean, are exclusively colonized by *M. sacchari*. The Nearctic zone is colonized by both species, as a result of the recent introduction of *M. sorghi* in the Americas. In the Neotropical zone, *M. sacchari* was the only species observed before 2016 [12]. However, EF1-alpha sequences obtained from three samples collected in 2020 (M. Kuki and C. Menezes, personal communication) show that *M. sorghi* is now present in Brazil. In Asia, *M. sorghi* is present in China and India, and *M. sacchari* in Cambodia. In Australia and in Hawaii, only *M. sacchari* was detected.

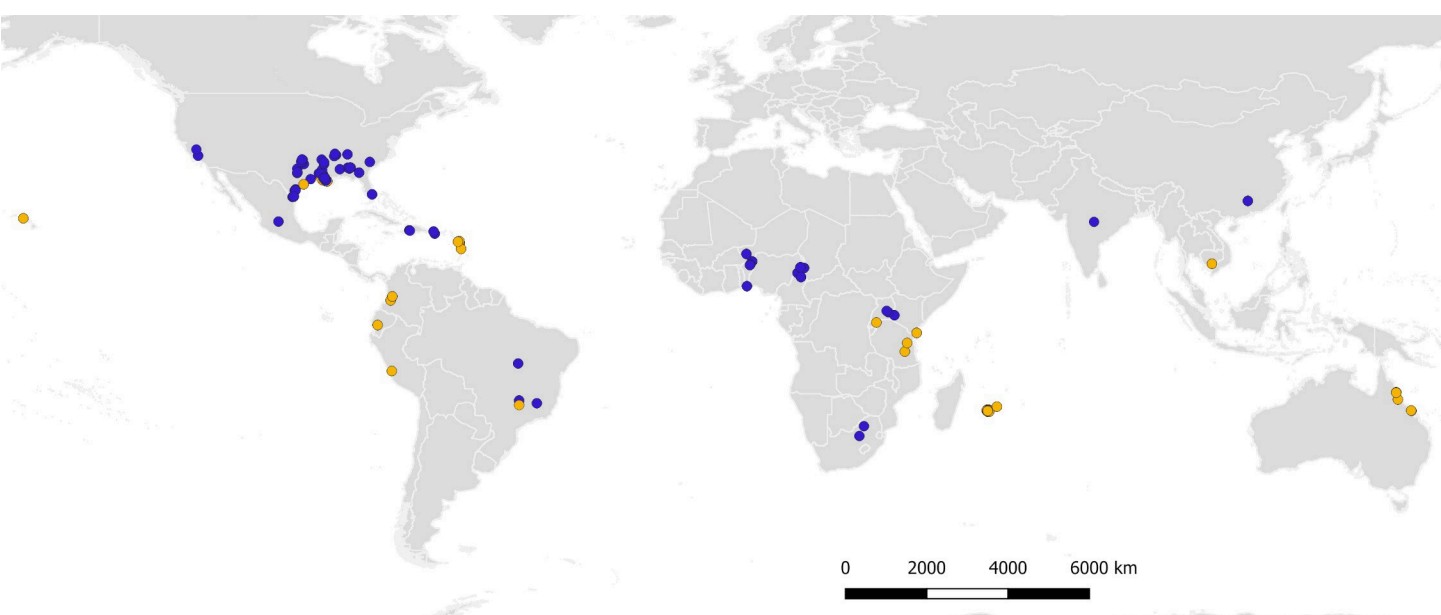

**Fig 6. Molecular identification with SSRs and sequencing of COI or EF1-α of 2,332 specimens: Blue = *M. sorghi*, orange = *M. sacchari*.** Data from India is the EF1-α sequence Genbank accession KU048048.1 (exact geographical location within India not available). *M. sorghi* data from Brazil are EF1- α sequences (M. Kuki and C. Menezes personal communication). The map was drawn using QGIS 3.4 (www.qgis.org). The maps of the administrative boundaries of the countries and states was uploaded from the database of Global Administrative Areas GADM 3.4 (www.gadm.org).

## Discussion

Genetic analyses with SSRs and three gene sequences showed that two genetic clades exist, one grouping MLL-A-E-F and the other grouping MLL-C-D. The multivariate morphometric data analysis separated the specimens in two groups matching the two genetic clades. Comparison with a *M. sorghi* paratype and comparison of the *HindTibia*:*pt* ratio [6, 7, 14] confirmed that the genetic clade grouping MLL-A-F is *M. sorghi* while the clade grouping MLL-C-D is *M. sacchari*. The status of MLL-B (Australian specimens) remains to be confirmed: it is assigned to *M. sorghi* by SSRs and EF1-α sequence, but we did not confirm this assignment by morphometric means, due to a lack of specimens. Similarly, the lack of specimens prevented us from studying the morphology of MLL-E (all from China) and its taxonomic status remains uncertain, although SSRs and EF1-α sequence data both suggest it belongs to *M. sorghi*.

Three morphological criteria are useful for species separation: *pt*:*cauda*, *pt*:*siph* and *HindTibia*:*pt*. However, as observed by Blackman and Eastop [7, 14], there are no clear limits between species and values overlap largely (Fig 2). The ranges for *M. sacchari* vs. *M. sorghi* are respectively *pt*:*cauda* (1.72–2.35) vs. (2.19–2.8), *HindTibia*:*pt* (1.8–2.31) vs. (1.54–1.93) and *pt*:*siph* (2.71–4.28) vs. (2.26–3.19). Due to overlap, the use of these morphological criteria can lead to ambiguous results and should be applied quantitatively (i.e. at the population level) rather than qualitatively (i.e. at the individual level).

The molecular diagnostic methods for separation of *M. sacchari* and *M. sorghi* are summarized in Figs 7 and 8. COI can allow for the identification of *M. sacchari* through two specific SNPs in positions 263 or 294. But, because haplotype H1 (Table 4) is shared by both species, an unambiguous identification of *M. sorghi* with COI is sometimes not possible. With EF1-α, a specific SNP in position 637 allows the separation of both species. Genotyping with a single

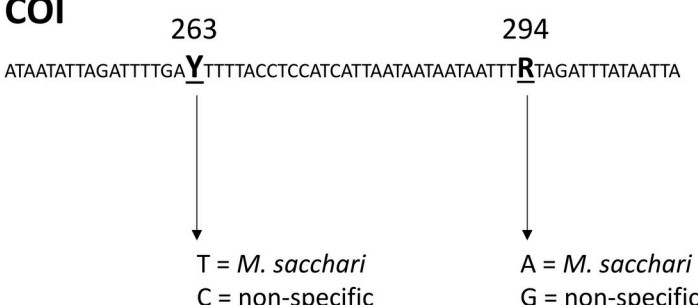

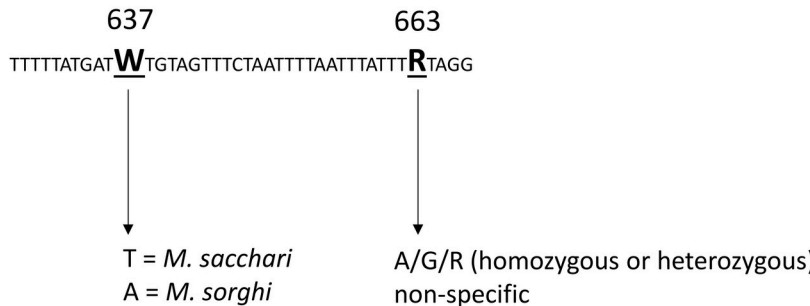

**Fig 7. Molecular diagnosis for separation of *M. sacchari* and *M. sorghi* using sequencing of COI or EF1-α.**

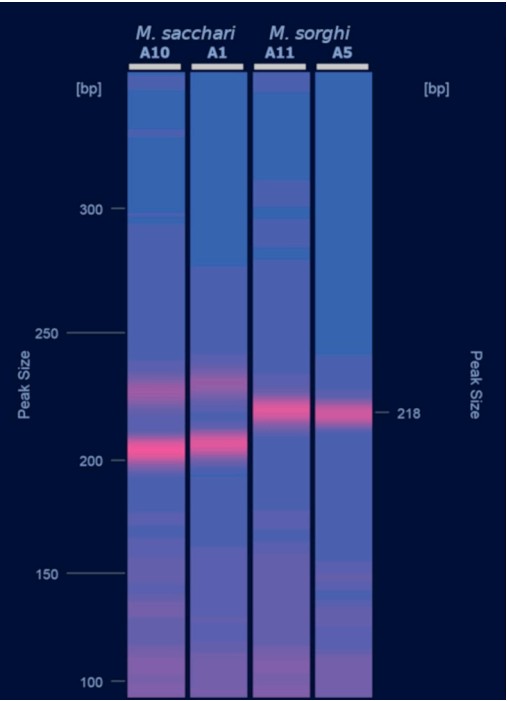

**Fig 8. Molecular diagnosis for separation of *M. sacchari* and *M. sorghi* using the SSR locus CIR-Ms-G01.** *M. sacchari* (MLL-D) are in lanes A10 (voucher # SNIB00040_0101) and A1 (voucher # SNIB00233_0102). *M. sorghi* (MLL-F) are in lanes A11 (voucher # SNIB00075_0101) and A5 (voucher # SNIB000237_0102). The PCR was carried out according to [14]. The migration of PCR products was carried out on a Qiagen Qiaxcel electrophoresis analyzer. The image was generated by the Qiaxcel ScreenGel 1.6.0 software (see original image in S1 Raw image).

SSR locus also allows the separation of both species. For example, the SSR locus CIR-Ms-G01 can be used for this purpose (Fig 8). The two alleles present at this locus in *M. sacchari* are separated by 25 to 31 bp, whereas *M. sorghi* genotypes are mostly homozygous or exhibit two alleles separated by 4 to 8 bp only (S6 Table).

We observed low genetic distances between *M. sacchari* and *M. sorghi* when comparing gene sequences that are widely used in aphid taxonomy. A 'borderline' distance between species in the COI barcode region is sometimes used by some authors to confirm species limits [32, 33]. However, it is now well recognized that there is no universal genetic distance separating aphid species and that low genetic difference between species can occur in aphids. Multiple examples of low COI, COII, CytB pairwise distance between species exist [34–36]. For example, a situation very similar to ours occurs in the genus *Megoura*, where Kim and Lee [37] observed an absence of COI, COII, CytB, ITS1, ITS2 divergence between *Megoura litoralis* Müller and *Megoura viciae* Buckton, which differ by a 0.2% P-distance on EF1-α.

According to Blackman et al. [6], *M. sorghi* is more likely observed on sorghum and *M. sacchari* more likely on sugarcane, although not absolutely specific to the hosts indicated by their names. In our study, the samples were distributed as follows: 14 from sugarcane vs. 94 from sorghum (all *Sorghum* species together) for *M. sorghi*, and 18 samples from sorghum vs. 69 from sugarcane for *M. sacchari*. Although our sampling plan was not designed to test host plant preference, the difference of preference between the two species appears obvious. The results of Boukari et al. [38] obtained in Florida confirm this preference, showing that sugarcane harbors almost only *M. sacchari* (COI haplotypes H2 and H3), which is absent from aphids on *Sorghum* spp. The recent work by Paudyal et al. [39] in the USA also supports this

apparent preference. Indeed, using host transfer experiments, these authors have demonstrated that MLL-F strains collected from *Sorghum* spp. exhibited a higher fitness on sorghum than on sugarcane, and that an MLL-D strain collected from sugarcane exhibited a higher fitness on sugarcane than on sorghum. According to this host preference difference, we suggest that the common name 'sugarcane aphid' should be used for *M. sacchari* and 'sorghum aphid' for *M. sorghi*.

Our study shows that the invasive genotype responsible for outbreaks on sorghum in North and Central America and the Caribbean islands since 2013 is MLL-F [12], which, belongs to *M. sorghi*, while the genotype present before 2013 (MLL-D) is *M. sacchari*. If the hypothesis of a lower fitness of *M. sacchari* on sorghum compared to *M. sorghi* is confirmed, this would explain why no damage was observed on sorghum prior to the introduction of *M. sorghi* to the Americas.

## Supporting information

**S1 Fig. *Melanaphis* apterous female habitus.**
(PDF)

**S2 Fig. Evanno method inferring k = 2.**
(PDF)

**S1 Raw image. Original image generated by the Qiaxcel ScreenGel software used to draw Fig 8.**
(PDF)

**S1 Table. List of samples and specimens.**
(XLSX)

**S2 Table. Comparison of morphological characteristics of *M. sacchari* and *M. sorghi* viviparous alate females.**
(PDF)

**S3 Table. CO1 haplotypes, position and nature of nucleotide substitutions.**
(PDF)

**S4 Table. CO2 haplotypes, position and nature of nucleotide substitutions.**
(PDF)

**S5 Table. EF1-α haplotypes, position and nature of nucleotide substitutions.**
(PDF)

**S6 Table. Observed microsatellite Multi Locus Genotypes (MLG).**
(PDF)

## Acknowledgments

We are grateful to Hughes Telismart, Magali Hoarau and Antoine Franck for technical assistance. We also thank Susan E. Halbert for her helpful comments on this manuscript. We also acknowledge Armelle Coeur d'Acier (CBGP), Laurent Fauvre and Thierry Bourgoin (MNHN) for providing access to the CBGP and MNHN collections. We also acknowledge M. Kuki (Innovative Seed Solutions, Brazil) and C. Menezes (Embrapa, Brazil) for providing us EF1-alpha and COI sequences from Brazil specimens.

## Author Contributions

**Conceptualization:** Samuel Nibouche, Laurent Costet, Roger L. Blackman.

**Data curation:** Samuel Nibouche, Laurent Costet, Joëlle Sadeyen.

**Formal analysis:** Samuel Nibouche, Anne-Sophie Zoogones.

**Funding acquisition:** Samuel Nibouche.

**Investigation:** Samuel Nibouche, Laurent Costet, Joëlle Sadeyen, Anne-Sophie Zoogones.

**Methodology:** Samuel Nibouche, Laurent Costet, Joëlle Sadeyen.

**Project administration:** Samuel Nibouche.

**Resources:** Samuel Nibouche, Raul F. Medina, Jocelyn R. Holt, Joëlle Sadeyen, Paul Brown.

**Supervision:** Samuel Nibouche, Laurent Costet, Joëlle Sadeyen.

**Validation:** Samuel Nibouche, Laurent Costet, Joëlle Sadeyen.

**Visualization:** Samuel Nibouche.

**Writing – original draft:** Samuel Nibouche, Laurent Costet, Raul F. Medina, Jocelyn R. Holt, Joëlle Sadeyen, Roger L. Blackman.

**Writing – review & editing:** Samuel Nibouche, Laurent Costet, Raul F. Medina, Jocelyn R. Holt, Joëlle Sadeyen, Anne-Sophie Zoogones, Paul Brown, Roger L. Blackman.

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
