## [Decision Letter · Decision Letter 0]

19 Jan 2021

PONE-D-20-32992

Morphometric and Molecular Discrimination of the sugarcane aphid, Melanaphis sacchari, (Zehntner, 1897) and the sorghum aphid Melanaphis sorghi (Theobald, 1904)

PLOS ONE

Dear Dr. Nibouche,

Thank you for submitting your manuscript to PLOS ONE. After careful consideration, we feel that it has merit but does not fully meet PLOS ONE’s publication criteria as it currently stands. Therefore, we invite you to submit a revised version of the manuscript that addresses the points raised during the review process.

We look forward to receiving your revised manuscript.

Kind regards,

Bernd Schierwater, Ph.D

Academic Editor

PLOS ONE

Journal Requirements:

4. We note that Figure 6 in your submission contain map images which may be copyrighted. All PLOS content is published under the Creative Commons Attribution License (CC BY 4.0), which means that the manuscript, images, and Supporting Information files will be freely available online, and any third party is permitted to access, download, copy, distribute, and use these materials in any way, even commercially, with proper attribution. For these reasons, we cannot publish previously copyrighted maps or satellite images created using proprietary data, such as Google software (Google Maps, Street View, and Earth). For more information, see our copyright guidelines: http://journals.plos.org/plosone/s/licenses-and-copyright.

4.1.    You may seek permission from the original copyright holder of Figure 6 to publish the content specifically under the CC BY 4.0 license. 

4.2.    If you are unable to obtain permission from the original copyright holder to publish these figures under the CC BY 4.0 license or if the copyright holder’s requirements are incompatible with the CC BY 4.0 license, please either i) remove the figure or ii) supply a replacement figure that complies with the CC BY 4.0 license. Please check copyright information on all replacement figures and update the figure caption with source information. If applicable, please specify in the figure caption text when a figure is similar but not identical to the original image and is therefore for illustrative purposes only.

Reviewers' comments:

Reviewer's Responses to Questions

**Comments to the Author**

1. Is the manuscript technically sound, and do the data support the conclusions?

Reviewer #1: Yes

Reviewer #2: Yes

2. Has the statistical analysis been performed appropriately and rigorously? 

Reviewer #1: Yes

Reviewer #2: Yes

3. Have the authors made all data underlying the findings in their manuscript fully available?

Reviewer #1: Yes

Reviewer #2: Yes

4. Is the manuscript presented in an intelligible fashion and written in standard English?

Reviewer #1: Yes

Reviewer #2: Yes

5. Review Comments to the Author

Reviewer #1: The preseneted manuscript is an excellent example of combine morphological/molecular study resolving the problem of identification of the two important pest species of aphids. The Material and Method section is fully and precisly decsribed. The introduction and the discussion are adequate. My only advise is to change the order in the Results section as should be started with the molecular ones as it was described in the M&M. In the Table S1 in the locality the United States should be added as now are listed countries and states. S2 Table should be included into the main manuscript as there are important data concerning identification of the studied species.

Reviewer #2: The discrimination study of sugarcane aphid was performed with great care and up to date methods and statistics. The manuscript is well written and can easily be understood. The observations are interesting and important for our community.

The results show the importance of using more than one marker or taxonomic indicator for species identification or discrimination, which in my personal opinion should be a standard procedure! I congratulate the authors for a well done study and recommend publication.

6. PLOS authors have the option to publish the peer review history of their article (what does this mean?). If published, this will include your full peer review and any attached files.

Reviewer #1: **Yes: **Karina Wieczorek

Reviewer #2: No

---

## [Author Response · Author response to Decision Letter 0]

4 Mar 2021

Reviewer #1: The presented manuscript is an excellent example of combine morphological/molecular study resolving the problem of identification of the two important pest species of aphids. The Material and Method section is fully and precisely described. The introduction and the discussion are adequate. 

My only advice is to change the order in the Results section as should be started with the molecular ones as it was described in the M&M. 

-> We agree that M&M and Results should follow the same order. We have preferred to change the order of the M&M section to correspond to that of Results section (this modification is not in track change mode so as not to disturb readability).

In the Table S1 in the locality the United States should be added as now are listed countries and states. 

-> We have added these data in the S1 Table.

S2 Table should be included into the main manuscript as there are important data concerning identification of the studied species.

-> We have moved S2 Table to the Results section, as Table 2 (this modification is not in track change mode so as not to disturb readability).

Reviewer #2: The discrimination study of sugarcane aphid was performed with great care and up to date methods and statistics. The manuscript is well written and can easily be understood. The observations are interesting and important for our community.

The results show the importance of using more than one marker or taxonomic indicator for species identification or discrimination, which in my personal opinion should be a standard procedure! I congratulate the authors for a well done study and recommend publication.

-> No modification requested.

---

## [Editor Report · Decision Letter 1]

8 Mar 2021

Morphometric and Molecular Discrimination of the sugarcane aphid, Melanaphis sacchari, (Zehntner, 1897) and the sorghum aphid Melanaphis sorghi (Theobald, 1904)

PONE-D-20-32992R1

Dear Dr. Nibouche,

We’re pleased to inform you that your manuscript has been judged scientifically suitable for publication and will be formally accepted for publication once it meets all outstanding technical requirements.

Kind regards,

Bernd Schierwater, Ph.D

Academic Editor

PLOS ONE
---

## [Editor Report · Acceptance letter]

16 Mar 2021

PONE-D-20-32992R1 

Morphometric and molecular discrimination of the sugarcane aphid, *Melanaphis sacchari*, (Zehntner, 1897) and the sorghum aphid *Melanaphis sorghi* (Theobald, 1904) 

Dear Dr. Nibouche:

I'm pleased to inform you that your manuscript has been deemed suitable for publication in PLOS ONE. Congratulations! Your manuscript is now with our production department. 

Kind regards, 

on behalf of

Prof. Bernd Schierwater 

Academic Editor

PLOS ONE